

# Rapid evolution of the *Helicobacter pylori* AlpA adhesin in a high gastric cancer risk region from Colombia

Andrés Julián Gutiérrez-Escobar[1,2], Gina Méndez-Callejas[1], Orlando Acevedo[3] and Maria Mercedes Bravo[4]

[1] Grupo de Investigaciones Biomédicas y Genética Humana Aplicada—GIBGA, Programa de medicina, Universidad de Ciencias Aplicadas y Ambientales U.D.C.A., Bogotá, Colombia
[2] Doctorado en Ciencias Biológicas, Pontificia Universidad Javeriana, Bogotá, Colombia
[3] Grupo de Biofísica y Bioquímica Estructural, Facultad de Ciencias, Pontifica Universidad Javeriana, Bogotá, Colombia
[4] Grupo de Investigación en Biología del Cáncer, Instituto Nacional de Cancerología de Colombia, Bogotá, Colombia

## ABSTRACT

To be able to survive, *Helicobacter pylori* must adhere to the gastric epithelial cells of its human host. For this purpose, the bacterium employs an array of adhesins, for example, AlpA. The adhesin AlpA has been proposed as a major adhesin because of its critical role in human stomach colonization. Therefore, understanding how AlpA evolved could be important for the development of new diagnostic strategies. However, the genetic variation and microevolutionary patterns of *alpA* have not been described in Colombia. The study aim was to describe the variation patterns and microevolutionary process of *alpA* in Colombian clinical isolates of *H. pylori*. The existing polymorphisms, which are deviations from the neutral model of molecular evolution, and the genetic differentiation of the *alpA* gene from Colombian clinical isolates of *H. pylori* were determined. The analysis shows that gene conversion and purifying selection have shaped the evolution of three different variants of *alpA* in Colombia.

## INTRODUCTION

*Helicobacter pylori*, a Gram-negative bacterium, has persistently colonized the stomach of half of the human population (*Perez-Perez, Rothenbacher & Brenner, 2004*; *Khalifa, Sharaf & Aziz, 2010*). This infection produces an asymptomatic inflammation of the gastric epithelium, but in some patients, it progresses toward a more severe clinical disease, such as ulcers and gastric cancer (*Yakirevich & Resnick, 2013*).

Gastric cancer is the fifth most common cancer worldwide (*Jemal et al., 2010*; *Bertuccio et al., 2009*; *Forman & Sierra, 2014*), and it is the second leading cause of cancer deaths (*Ferlay et al., 2013*); an infection with *H. pylori* is the strongest factor risk for its development (*Helicobacter and Cancer Collaborative Group, 2001*). In Colombia, the prevalence of this infection is universally high (*Matta et al., 2017*). However, the gastric cancer risk increases

Corresponding author
Andrés Julián Gutiérrez-Escobar,
andregutierrez@udca.edu.co,
andresjulian1981@gmail.com

along with the altitudinal gradient (*Torres et al., 2013*). Thus, it is higher in the Andes region than along the Pacific coast (*Kodaman et al., 2014*); this phenomenon has been called the Colombian enigma (*Correa & Piazuelo, 2010*).

The bacterium has coevolved with its human host for a period of 60,000 years, since the first major migratory wave outside Africa (*Linz et al., 2007*). The bacterium has followed a similar dispersal pattern as its human host. Currently, seven populations of *H. pylori* have been identified: hpEurope, hpNEAfrica, hpAfrica1, hpAfrica2, hpAsia2, hpSahul and hpEastAsia (*Yamaoka, 2009*; *Falush et al., 2003*; *Achtman et al., 1999*; *Moodley et al., 2009*; *Devi et al., 2007*). A recent human migratory event was the colonization of the Americas 525 years ago. During this colonization, new pathogens arrived to the American continent, including new strains of *H. pylori,* which caused the disappearance of 80% of the native population (*Bianchine & Russo, 1992*; *Parrish et al., 2008*). It has been reported that *H. pylori* has followed unique evolutionary pathways in Latin-America (*Gutiérrez-Escobar et al., 2017*; *Muñoz Ramírez et al., 2017*) and that the strains followed rapid adaptive processes in different countries of the region (*Thorell et al., 2017*), establishing local independent lineages.

The bacterium has virulence factors that correlate with the risk of developing gastric diseases (*Parsonnet et al., 1997*; *Nomura et al., 2002*; *Mahdavi et al., 2002*; *Yamaoka et al., 2002a*; *Yamaoka et al., 2002b*). *H. pylori* attach to the gastric epithelium thorough adhesins that contribute to the initial steps of the infection (*Matsuo, Kido & Yamaoka, 2017*). AlpA (∼56 kDa) is an adhesin that is encoded by the locus *alpAB* (*Alm et al., 2000*), which is essential for adherence to the human gastric epithelium (*Odenbreit et al., 1999*). This adhesin is expressed by all clinical isolates (*Odenbreit et al., 2009*) and is recognized in sera from infected patients (*Xue et al., 2005*), and in addition, it induces the secretion of interleukin-8 (IL-8) (*Lu et al., 2007*). AlpA represents an emerging virulence factor of *H. pylori* that has been gaining attention because of its potential as a vaccine target (*Sun et al., 2010*).

The aim of the present study was to describe the genetic diversity and microevolution of the adhesin AlpA at the population level in the high gastric cancer risk region of Colombia. Population genetics statistics and phylogenetic methods were performed to compare *H. pylori* strains from Colombian clinical isolates against strains from different geographical backgrounds.

## MATERIALS AND METHODS

### DNA and protein sequences

The DNA (*alpA*) and protein sequences (AlpA) were obtained from 115 genomes from the *H. pylori* stock collection, which was previously sequenced by our research group at the Instituto Nacional de Cancerología in Bogotá. The isolates belonged to patients with different types of gastric pathologies associated with *H. pylori* infection, as follows: 30 cases of Gastritis (G), 20 of Gastric Adenocarcinoma (GA), 28 of Atrophic Gastritis (AG), 30 of Intestinal Metaplasia, five of Gastritis concomitant with Duodenal Ulcer (G-DU) and two of Intestinal Metaplasia concomitant with Duodenal Ulcer (IM-DU) (*Gutiérrez-Escobar et al., 2017*).

The reference pool sequences were obtained from 34 *H. pylori* strains, as follows: HspAmerind: Cuz20, PeCan4, Puno135, Sat464, Shi112, Shi169, Shi417, Shi470, v225d; HpEurope: 26695, B8, G27, HPAG1, ELS37, Lithuania75, SJM180; HspAsia: F57, XZ274, 51, 52, 35A, F32, F16, F30, 83; HspAsia2: SNT49, India7; HspWestAfrica: J99, 908, PeCan18, 2017, 2018, Gambia94/24 and HpSouthAfrica: SouthAfrica7.

## Phylogenetic analysis of *alpA*

A total of 142 protein sequences for AlpA (108 sequences from Colombian isolates and 34 from reference) were aligned using the software Muscle V 3.8.31 (*Edgar, 2004*); the evolutionary model and the phylogenetic reconstruction was determined using MEGA V 7 (*Kumar, Stecher & Tamura, 2016*) and the NJ algorithm (*Saitou & Nei, 1987*) with 1,000 bootstrap repetitions for statistical robustness.

## Genetic diversity, natural selection tests and differentiation analysis of *alpA*

The following population statistics: number of haplotypes ($H$), haplotype diversity ($Hd$), nucleotide diversity ($Pi$), average number of nucleotide differences ($k$), Theta estimator ($\theta w$) and recombination events (Rm) analyses were performed using DnaSP v 5.10 (*Librado & Rozas, 2009*). Deviations of the neutral model of molecular evolution were tested using the Tajima test (*Tajima, 1989*) and the $Z$-test in which the average number of synonymous substitutions per synonymous site (dS) and the average number of non-synonymous substitutions per non-synonymous site (dN) were calculated using the modified Nei-Gojobori method with the Junkes-Cantor correction. The variance of the difference was computed using the bootstrap method (1,000 replicates) using Mega v 7 (*Kumar, Stecher & Tamura, 2016*). Finally, a sliding window analysis was applied to detect the evolutionary rate $\omega = dN/dS$ throughout the gene to identify specific regions under natural selection using the software DnaSP v 5.10 (*Librado & Rozas, 2009*).

The DNA sequences of Colombian isolates were grouped according to the gastric pathology of the patients: Gastritis (G), Gastric Adenocarcinoma (GA), Atrophic Gastritis (AG), Intestinal Metaplasia (IM), Gastritis concomitant with Duodenal Ulcer (GDU) and Intestinal Metaplasia concomitant with Duodenal Ulcer (IM-DU); the reference sequences were grouped according to their geographic origin. To detect the genetic heterogeneity and genetic flow, the following tests were applied: Hst, Kst, Kst*, Z, Z*. The HBK, Snn and chi squared tests were performed using the haplotype frequencies under the permutation of 1,000 repetitions, as well as the tests for the haplotype diversity of Gst, Nst, Fst, and Da, and the gene flow (Nm), a measure of the genetic interaction, was estimated from FsT (*Slatkin, 1985*; *Slatkin, 1987*) using the software DnaSP v 5.10 (*Librado & Rozas, 2009*).

## Gene conversion analysis of *alpA*

To test whether gene conversion generates genetic diversity in the *alpA,* the Betran's method (*Betran et al., 1997*) implemented in the DnaSP v 5.10 software was used to compare Colombian isolates and the reference pool populations. RDP3 v 3.4 software was used to test gene conversion in the overall population using the GENECONV algorithm (*Sawyer, 1989*). Only conversion tracks with $p < 0.05$ were considered.

## Type I functional divergence and site specific positive selection analyses

The 3D structure of the AlpA protein was predicted using the server I-TASSER (*Yang et al., 2015*) and tested using the MetaMQAPII server (*Pawlowski et al., 2008*). DIVERGE V 3 software (*Gu et al., 2013*) was used to estimate type-I functional divergence, which detects functional changes in a protein based on site-specific shifts in the evolutionary rates (*Gu, 1999*). The software tested whether a significant change in the evolution rate has occurred by calculating the coefficient of divergence ($\theta D$). Positive and negative selection was evaluated as the proportions of synonymous to non-synonymous substitution rates. The DNA sequences were aligned using the software Muscle V 3.8.31 (*Edgar, 2004*), and the alignment file was pre-processing by screening for recombination breakpoints using the GARD algorithm implemented by the HyPhy software (*Kosakovsky Pond et al., 2006a*; *Kosakovsky Pond et al., 2006b*). Then, the processed file was tested for selection using the FEL and IFEL algorithms using the datamonkey server (*Kosakovsky Pond & Frost, 2005*). Episodic diversifying selection was detected using the MEME algorithm (*Murrell et al., 2012*). A $p < 0.01$ was considered to be statistically significant for all of the selection tests. The RELAX test to detect relaxed selection on the codon-based phylogenetic framework of *alpA* from Colombian isolates was performed using the datamonkey server (*Kosakovsky Pond & Frost, 2005*).

## RESULTS

A total of 142 sequences were used in this study: 86 were obtained from Colombian patients with different gastric diseases, and 34 sequences were obtained from reference strains from GenBank. The phylogenetic tree of AlpA using only the reference pool showed three clades: one clustering sequences from HspAmerind and HspAsia populations, the second clustering sequences from HpEurope and HspSouthIndia, and the last clustering sequences exclusively from HspWestAfrica. The tree does not reflect a clear separation between HspAmerind and HspAsia nor between HpEurope and HspSouthIndia, which suggests that recombination and gene conversion gave rise to its evolutionary pattern (Fig. 1).

The phylogenetic tree including Colombian isolates showed an intricate pattern of clades. Five major clades were detected: (1) cluster sequences from HspWestAfrica, HspColombia and HpEurope; (2) cluster sequences from HpEurope, HspSouthIndia and HspColombia; (3) cluster sequences from HspColombia; (4) cluster sequences from HspAsia, HspAmerind; and (5) cluster sequences from HpEurope, HspAsia and HspColombia (Fig. 2). The phylogenetic tree of the Colombian isolates showed three major clades, called Col1, Col2 and Col3; a more detailed analysis allowed us to show that the phylogenetic tree had seven subclades, called 1 to 7, which indicates that an intense evolutionary process is taking place between Colombian isolates with respect to AlpA (Fig. 3).

The analysis of the nucleotide diversity (2.5-fold) and the average number of nucleotide differences (3.7-fold) showed them to be higher in the reference pool than in the Colombian

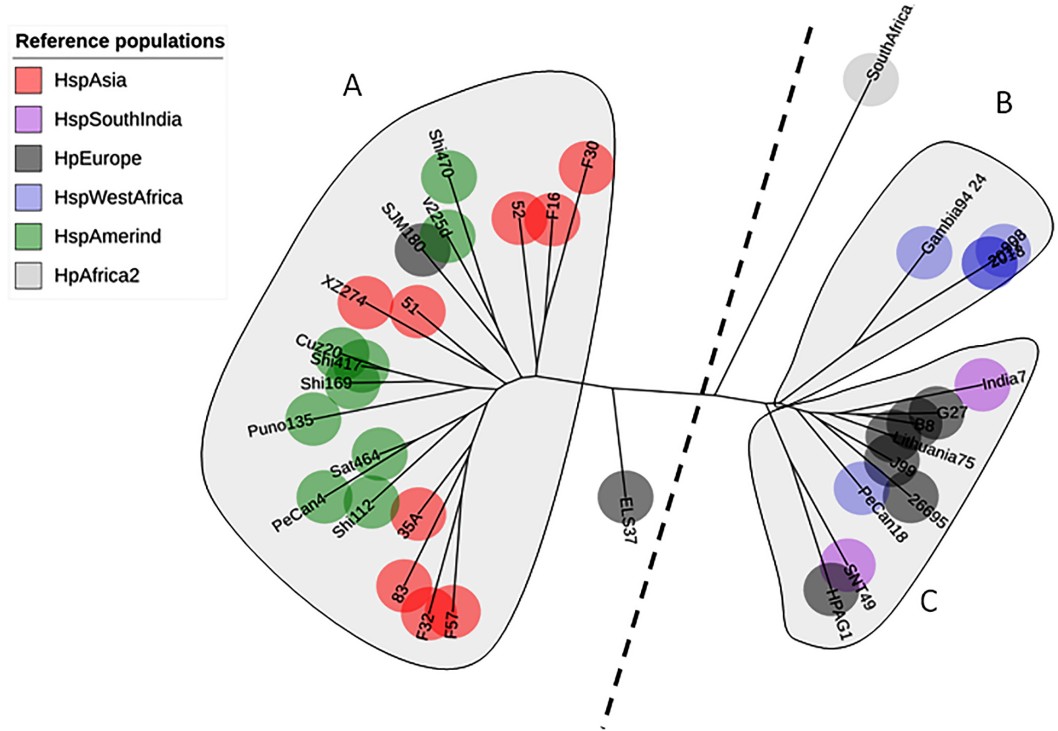

**Figure 1  Phylogenetic tree of AlpA from reference strains of *H. pylori*.** The evolutionary history was inferred using the Neighbor-Joining method. The optimal tree with the sum of branch length = 0.84663616 is shown. The percentage of replicate trees in which the associated taxa clustered together in the bootstrap test (1,000 replicates); significant consensus tree branches are showed. The evolutionary distances were computed using the JTT matrix-based method and are in the units of the number of amino acid substitutions per site. The rate variation among sites was modeled with a gamma distribution (shape parameter = 2). The analysis involved 34 amino acid sequences. (A) Clade East. (B) Clade Western. All positions containing gaps and missing data were eliminated. There were a total of 451 positions in the final dataset.

isolates. Similarly, the theta estimator showed that the reference pool was significantly more diverse than the Colombian counterparts. The number of haplotypes was 3-fold higher in the Colombian isolates than that observed in the reference pool, but the haplotype diversity was similarly higher in both populations; the total number of haplotypes was 134, with an extreme value for the haplotype diversity. Recombination events were 1.3-fold higher in the Colombian isolates than in the reference pool (Table 1).

To test the deviations of the neutral model of molecular evolution, the Tajima and $Z$-test were employed. The Tajima's $D$ test was negative and the $Z$-test showed significant results for neutrality, indicating that in overall average the synonymous substitution rate was similar the non-synonymous substitution rate ($p = 0.005$) (Table 2). However, the sliding windows analysis showed that several regions of the *alpA* has an $\omega$ value above 1, which suggests that episodic positive diversifying selection is also shaping the microevolutionary patterns of this gene in Colombia (Fig. 4).
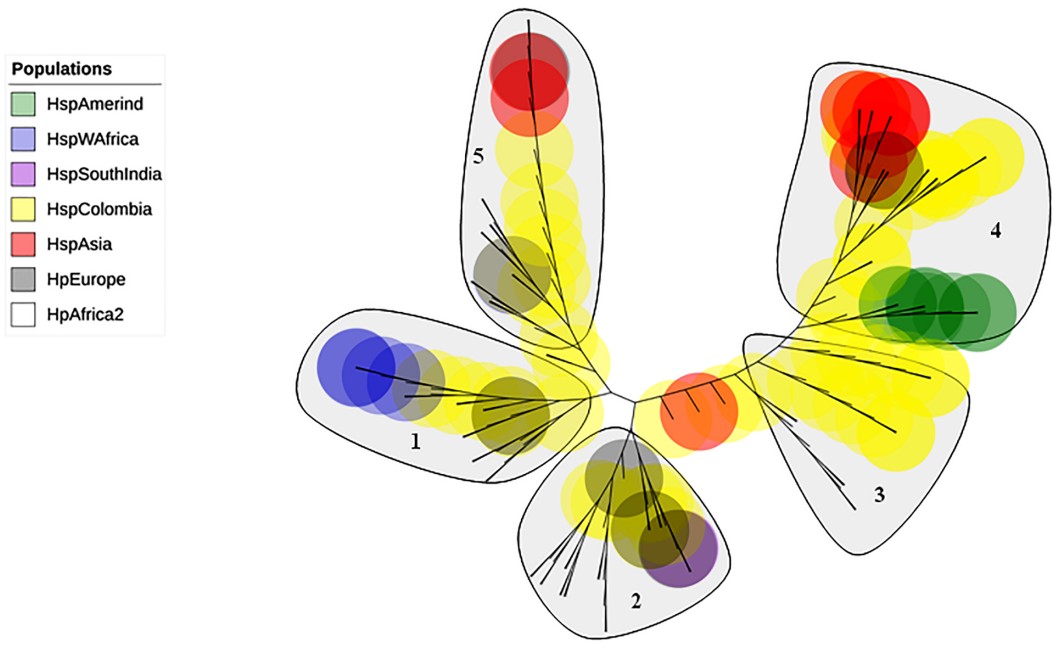

**Populations**
- HspAmerind
- HspWAfrica
- HspSouthIndia
- HspColombia
- HspAsia
- HpEurope
- HpAfrica2

**Figure 2** **Phylogenetic tree of AlpA proteins of *Helicobacter pylori*.** The evolutionary history was inferred using the Neighbor-Joining method. The optimal tree with the sum of branch length = 0.84663616 is shown. The percentage of replicate trees in which the associated taxa clustered together in the bootstrap test (1,000 replicates); significant consensus tree branches are showed. The evolutionary distances were computed using the JTT matrix-based method and are in the units of the number of amino acid substitutions per site. The rate variation among sites was modeled with a gamma distribution (shape parameter = 2). The analysis involved 142 amino acid sequences. All positions containing gaps and missing data were eliminated. There were a total of 451 positions in the final dataset. Five major clades were detected: (1) Cluster sequences from HspWestAfrica, HspColombia and HpEurope; (2) Cluster sequences from HpEurope, HspSouthIndia and HspColombia; (3) Cluster sequences from HspColombia; (4) Cluster sequences from HpAsia, HspAmerind and HspColombia; and (5) Cluster sequences from HpEurope, HpAsia and HspColombia.

The test of genetic differentiation showed that the Colombian population was well-differentiated from the reference pool, but as expected, the Nm obtained from FsT was 3.63, which indicates a moderate gene flow between the Colombian isolates (*Slatkin, 1985*; *Slatkin, 1987*) (Table 3). To assess the isolation between pairs of populations, the Colombian isolates were organized into seven groups based on the histopathological diagnosis, and then, the *alpA* DNA sequences were compared pairwise with the reference pool clustered according to their phylogeographic classification. Population isolation was found between the Colombian populations and all subpopulations (Table 4). Two gene conversion algorithms were applied to the population to test whether this process was involved in the evolutionary pattern of *alpA*. Betran's algorithm found 54 conversion tracks between the sequences: 20.5% for the reference pool and 44.4% for the Colombian isolates, and the genconv algorithm found six tracks of gene conversion between the sequences that belong exclusively to the Colombian isolates (Fig. 5).

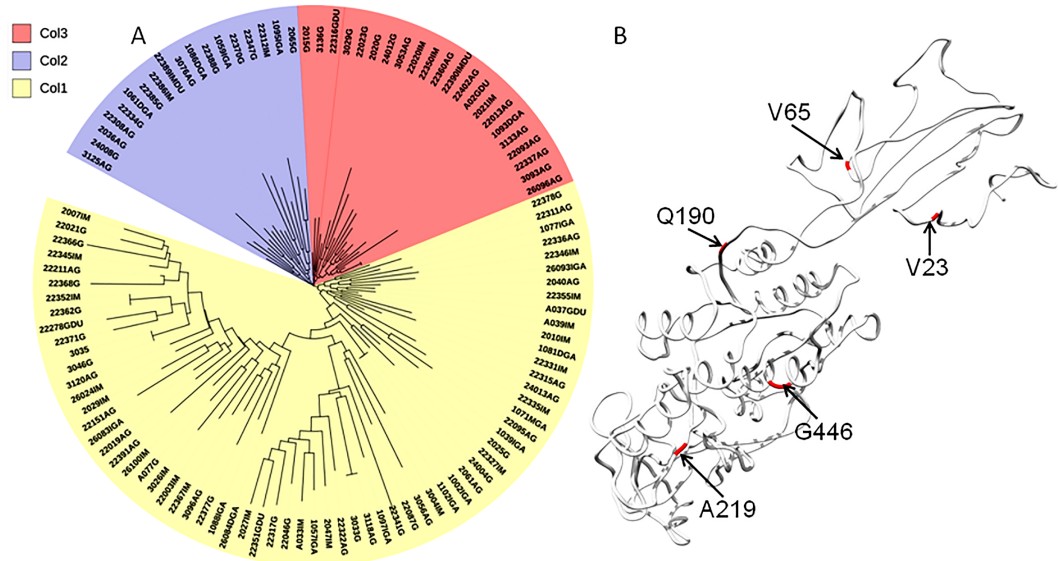

**Figure 3 Phylogenetic tree of AlpA proteins from Colombian isolates of *Helicobacter pylori*.** (A) The evolutionary history was inferred using the Neighbor-Joining method. The optimal tree with the sum of branch length = 0.84663616 is shown. The percentage of replicate trees in which the associated taxa clustered together in the bootstrap test (1,000 replicates); significant consensus tree branches are showed. The evolutionary distances were computed using the JTT matrix-based method and are in the units of the number of amino acid substitutions per site. The rate variation among sites was modeled with a gamma distribution (shape parameter = 2). The analysis involved 108 amino acid sequences. All positions containing gaps and missing data were eliminated. There were a total of 451 positions in the final dataset. Three major clades were detected: (1) Col1; (2) Col2 and (3) Col3. (B) Comparison all-vs-all between Colombian phylogenetic tree clusters reveals that Col1/Col3 clades have five sites with a θD > 0.8 indicating a strong signal of functional divergence.

**Table 1 Genetic diversity statistics for alpA.** The genetic diversity statistics were calculated by using 34 sequences from reference pool and 108 sequences from Colombian isolates of Helicobacter pylori.

| | | | Single nucleotide polymorphism | | | | | |
|---|---|---|---|---|---|---|---|---|
| *n* | Sites | Ss | *k* | *H* | *Hd* | θw | *π* | Rm |
| *Reference* | | | | | | | | |
| 34 | 1,324 | 774 | 200.868 | 33 | 0.998 | 0.143 | 0.152 | 70 |
| *Colombian* | | | | | | | | |
| 108 | 1,384 | 306 | 53.335 | 102 | 0.998 | 0.042 | 0.039 | 91 |
| *Colombian and references* | | | | | | | | |
| 142 | 1,223 | 754 | 81.411 | 134 | 0.999 | 0.112 | 0.067 | 98 |

**Notes.**

The estimators were presented in three groups: references, Colombian and both together.

*n*, number of isolates; *sites*, total number of sites analyzed (excluding gaps); *Ss*, number of segregant sites; *k*, average number of nucleotide differences by sequence pairs; *H*, number of haplotypes; *Hd*, Haplotype diversity; θw, Watterson estimator; *π*, nucleotide diversity per site and; RM, recombination events.

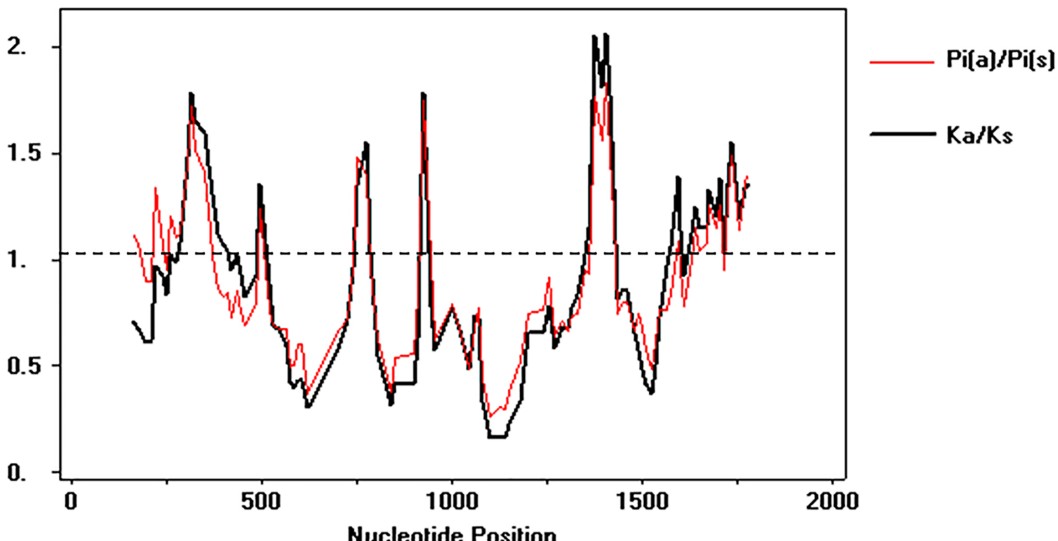

**Figure 4  Sliding windows analysis of $\omega$ rate**  Evolutionary rate $\omega = \mathrm{dN/dS}$ throughout the gene to identify specific regions under natural selection. The analysis involved 142 DNA sequences, 108 from Colombian isolates and 34 as phylogeographic references. The analysis was performed using DnaSP v 5.10.

**Table 2  Deviations of the neutral model of molecular evolution.** The analysis involved 142 nucleotide sequences. Tajima test: codon positions included were first. All positions containing gaps and missing data were eliminated.

| | | Tajima test | | | | | Codon Based $Z$-test of selection | | |
|---|---|---|---|---|---|---|---|---|---|
| $m$ | $S$ | $p_s$ | $\Theta$ | $\pi$ | $D$ | | $\mathrm{dN} = \mathrm{dS}$ | $\mathrm{dN} > \mathrm{dS}$ | $\mathrm{dN} < \mathrm{dS}$ |
| 142 | 217 | 0.604 | 0.109 | 0.059 | $-1.476$ | | $-2.878^{***}$ | $-0.213$ ns | 0.214 ns |

**Notes.**
There were a total of 359 positions in the final dataset, $m$, number of sequences, $n$, total number of sites; $S$, Number of segregating sites; $ps$, $S/n$; $\Theta$, ps/a1; $\pi$, nucleotide diversity, and $D$ is the Tajima statistic. $Z$-Test: Codon-based test of neutrality for analysis averaging over all sequence pairs. The probability of rejecting the null hypothesis of strict-neutrality ($\mathrm{dN} = \mathrm{dS}$) is shown. Values of $P$ less than 0.05 are considered significant at the 5% level and are highlighted ($^{***}p < 0.001$; ns, not significative). The test statistics: $\mathrm{dN} < \mathrm{dS}$, neutrality; $\mathrm{dN} > \mathrm{dS}$ positive selection; $\mathrm{dN} < \mathrm{dS}$ purifying selection are shown. dS and dN are the numbers of synonymous and nonsynonymous substitutions per site, respectively. The variance of the difference was computed using the bootstrap method (1,000 replicates). Analyses were conducted using the Nei-Gojobori method. All positions containing gaps and missing data were eliminated.

**Table 3  Genetic heterogeneity and genetic flow of *alpA*.** Genetic heterogeneity and genetic flow were detected using the following tests: Hst, Kst, Kst*, Z, Z*.

| | | Genetic differentiation | | | | | Genetic flow | | | |
|---|---|---|---|---|---|---|---|---|---|---|
| $Chi2$ | $Hst$ | $Kst$ | $Kst^*$ | $Z$ | $Z^*$ | $Snn$ | $Gst$ | $GammaST$ | $Nst$ | $Fst$ |
| 1501.4 ns | $0.003^{**}$ | $0.090^{**}$ | $0.034^{***}$ | $4011.8^{***}$ | $7.837^{***}$ | $0.316^{***}$ | 0.011 (44.82) | 0.178 (2.30) | 0.114 (3.88) | 0.120 (3.63) |

**Notes.**
The HBK, Snn and chi squared tests were performed using the haplotype frequencies under the permutation of 1,000 repetitions, as well as the tests for haplotype diversity of Gst, Nst, Fst. In parenthesis, the gene flow (Nm) value was estimated from FsT using the software DnaSP 5.10.
Statistical significance: $^*p < 0.05$. $^{**}p < 0.01$. $^{***}p < 0.001$. ns, not significative.

**Table 4** Pairwise analysis of differentiation and genetic flow in the populations of *alpA*.

| Populations Amerind | FsT | GammaST |
|---|---|---|
| G vs. hspAmerind | **0.256** | **0.113** |
| GA vs. hspAmerind | **0.325** | **0.151** |
| IM vs. hspAmerind | **0.276** | **0.141** |
| IGA vs. hspAmerind | **0.281** | **0.212** |
| DGA vs. hspAmerind | **0.379** | **0.328** |
| G/DU vs. hspAmerind | **0.246** | **0.243** |
| IM/DU vs. hspAmerind | **0.504** | **0.373** |
| **Europe** | | |
| G vs. HpEurope | **0.125** | **0.143** |
| GA vs. HpEurope | **0.125** | **0.148** |
| IM vs. HpEurope | **0.131** | **0.155** |
| IGA vs. HpEurope | **0.133** | **0.160** |
| DGA vs. HpEurope | **0.119** | **0.128** |
| G/DU vs. HpEurope | **0.11** | **0.123** |
| IM/DU vs. HpEurope | **0.153** | 0.093 |
| **Asia** | | |
| G vs. HpAsia | **0.119** | **0.123** |
| GA vs. HpAsia | **0.135** | **0.137** |
| IM vs. HpAsia | **0.123** | **0.132** |
| IGA vs. HpAsia | **0.132** | **0.133** |
| DGA vs. HpAsia | **0.153** | **0.118** |
| G/DU vs. HpAsia | **0.11** | 0.099 |
| IM/DU vs. HpAsia | **0.204** | 0.086 |
| **Africa** | | |
| G vs. HpAfrica | **0.128** | **0.134** |
| GA vs. HpAfrica | **0.123** | **0.137** |
| IM vs. HpAfrica | **0.131** | **0.148** |
| IGA vs. HpAfrica | **0.124** | **0.163** |
| DGA vs. HpAfrica | **0.099** | **0.138** |
| G/DU vs. HpAfrica | **0.111** | **0.145** |
| IM/DU vs. HpAfrica | **0.165** | **0.121** |

**Notes.**

The FsT and GammaST differentiation parameters are shown. The populations of Colombian isolates are denoted as follows: G, Gastritis; GA, Gastric Adenocarcinoma; IM, Intestinal Metaplasia; IGA, Intestinal Gastric Adenocarcinoma; DGA, Diffuse Gastric Adenocarcinoma; G/DU, Gastritis + Duodenal ulcer; IM/DU, Metaplasia + Duodenal ulcer. A total of 142 sequences were used for the analyses using DnaSP V 5.10. The significance of genetic differentiation was assumed as follow: <0.05 = little, 0.05–0.15 = moderate, 0.15–0.25 = high and >0.25 = complete (*Hartl & Clark, 1997*). Significative FsT values are highlighted in bold.

Functional divergence analysis among the AlpA proteins sequences based on the three major clades found in the phylogenetic tree of the Colombian isolates was performed using Gu's type-I method. The pairwise comparisons between the Col1/Col3 clades showed five sites with a $\theta D > 0.8$ (Fig. 3B), which indicates that the protein presents sites with different evolutionary rates. The analysis of positive and negative selection covered 540 codon sites of

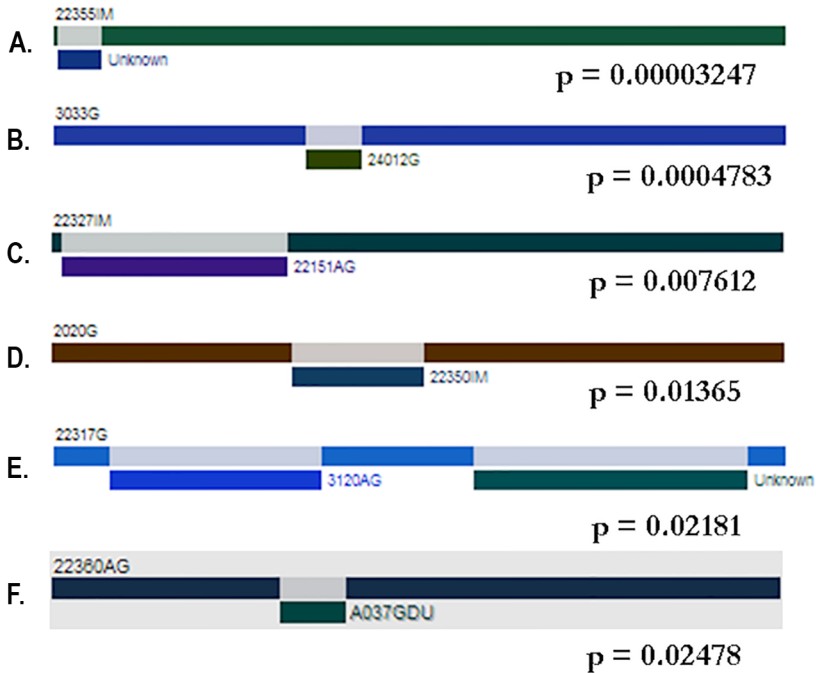

**Figure 5  Gene conversion tracks identified by the GENECONV method.** (A–F) are bars that represent genes. The colored rectangles represent the gene fragments or minor parents involved in a gene conversion event. The tracks were located predominantly at the 5′ region of the gene; however some events were also detected at 3′ region.

the AlpA protein. A similar number of sites were detected by the FEL and IFEL algorithms. FEL identified that 3.3% of the sites were under positive selection and 21.4% evolve under purifying selection. IFEL showed that 2.7% of the sites evolve under positive selection, with 12.2% under purifying selection, and finally, MEME showed that 5% of the adhesin was under episodic diversifying selection (Fig. 6). The relax test showed that when the internal branches were compared against the external branches, a significant pattern of natural selection intensification was detected ($K = 29.55$, $p = 5.48\text{e}^{-10}$, $LR = 38.50$).

## DISCUSSION

In this study, we describe the genetic diversity and microevolution of AlpA in Colombian isolates of *H. pylori* obtained from a high gastric cancer region composed of the cities of Bogotá, Tunja and the surrounding towns. The phylogenetic tree using the reference AlpA sequences revealed that the strains clustered according to their geographic origin; two main clades were observed: the Eastern and the Western clades. Then, within each main clade, there was no separation between the *H. pylori* subpopulations. This geographical segregation has been observed for other virulence factors of *H. pylori* (*Cao et al., 2005*; *Maeda et al., 1998*; *Oleastro et al., 2009a*; *Van Doorn et al., 1999*) and it is an indicative of the pre-colonization period. However, when the sequences of Colombian isolates are added to the analysis, the main East and Western clades fade, which indicates that the sequences share

A                                      B

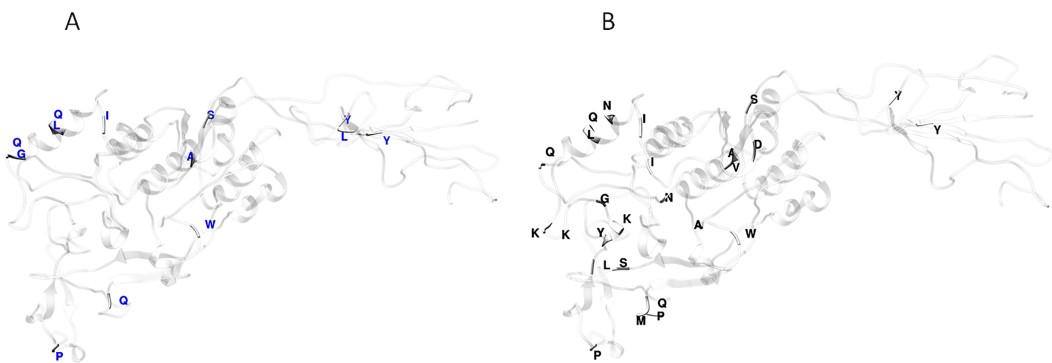

**Figure 6** **Protein model of AlpA showing the positive episodic diversified selected sites after the SBP correction thereby recombination.** (A) Positive selected sites were obtained from the tests FEL (18 sites) and IFEL (14 sites) showed only common sites at the protein structure in red. (B) Episodic diversifying selective sites were obtained using the test MEME (27 sites) in black, and both analyses were performed using the Datamonkey server from the codon alignment file. Only sites with a $p < 0.01$ are shown.

information by recombination and diversifying selection induced by the post-colonization period, when the host and bacterial populations mix together in Latin-America, and for the induction of different divergence rates between the paralogous/homologous members of different strains as a consequence of gene conversion (*Santoyo & Romero, 2005*).

The phylogenetic tree of Colombian isolates showed tree dominant clades entitled Col1, Col2 and Col3. The clades Col1 and Col2 have few branching events with a low number of members, but the Col3 clade showed five subclades. There was no association between the disease state and belonging to a specific clade ($p = 0.245$ Chi-square 2.81), which indicates that the *H. pylori* strains interchange DNA randomly between the Colombian strains. It is a well-established fact that the bacterium is naturally competent (*Dorer et al., 2013*).

The nucleotide diversity and the average number of nucleotide differences was lower in the Colombian *alpA* alleles that in the reference pool. However, the Colombian isolates showed a higher number of haplotypes and recombination events. The *alpA* gene from Colombian isolates has shown strong patterns of genetic differentiation, but it maintains the genetic flow that has produced new allelic variants of *alpA* in Colombia. The arrival of the HpEurope to Latin-America induced the replacement of the native HspAmerind population from urban zones (*Yamaoka et al., 2002a*; *Yamaoka et al., 2002b*) and perhaps produced a selective bottleneck. After the bottleneck, a subsequent population expansion emerged with the new recombinant *H. pylori* subtypes in Colombia. It has been proposed that recombination among *H. pylori* strains can induce the evolution of different subclones and genotypes (*Blaser & Berg, 2001*; *Blaser & Atherton, 2004*).

*H. pylori* has one of the highest mutation and recombination rates between bacterial species (*Suerbaum & Josenhans, 2007*). It has been identified that *H. pylori* has followed unique evolutionary pathways in Latin-America (*Muñoz Ramírez et al., 2017*), and the strains have followed rapid adaptive processes in different countries of the region (*Gutiérrez-Escobar et al., 2017*; *Thorell et al., 2017*). Another factor that contributes to the rapid evolution of the Colombian *H. pylori alpA* subtypes was the mestizo host. Perhaps

the immune and inflammatory responses of this new host and its genetic heterogeneity represented by the variability and distribution of receptors to which the bacterium can adhere could be a selective factor for the divergence among Colombian *H. pylori* strains (*Dubois et al., 1999*; *Oleastro et al., 2009b*). In fact, the host imposes a selective pressure that induces variation within the bacterium (*Thompson et al., 2004*).

Gene conversion was a major process that gave rise to the allelic variation of *alpA* in Colombia, and the paralogous interchange of the DNA fragments close to the 3′ region of the gene was considerable higher between the Colombian isolates than in the reference pool. In total, 60 recombination events were detected by the two applied algorithms, and despite the total number of sequences used in the analysis, only Colombian *alpA* alleles showed statistical signals of this type of recombination. Gene conversion has been identified previously in other *H. pylori* adhesins, for example, *homB* and *homA* (*Dorer et al., 2013*; *Dubois et al., 1999*; *Oleastro et al., 2009a*; *Oleastro et al., 2009b*; *Santoyo & Romero, 2005*; *Yamaoka et al., 2002a*; *Yamaoka et al., 2002b*), *sabB* and omp27 (*Talarico et al., 2012*) and *babA* (*Hennig, Allen & Cover, 2006*), and a Latin-American type of *babA* has also been recently reported (*Thorell et al., 2016*).

The interplay between positive selection and recombination has been detected in bacterial genomes (*Lefebure & Stanhope, 2007*; *Joseph et al., 2011*; *Orsi, Sun & Wiedmann, 2008*). The analysis of functional divergence analysis of the protein AlpA based on the comparison of the tree clades Col1, Col2 and Col3 showed that the protein residues 23V, 65V, 190Q, 219A and 446G have a significant evolutionary site variation rate with a θD value of 0.8, which confers functional differences between the members of the clades Col1/Col3. In Colombia, positive, episodic diversifying, purifying selection and recombination of *alpA* has given rise to the presence of rare alleles and new haplotypes in the emerging population, which have been maintained at the protein level by natural selection.

The action of natural selection might purge diversified genes from the population by strong purifying selection (*Lynch & Conery, 2000*) or lead to evolutionary novelties by positive selection (*Wertheim, Murrell & Smith, 2015*). When the internal branches of the phylogenetic tree of Colombian *alpA* alleles were compared against the external ones, we determined that there was a significant natural selection intensification operating in the *alpA* alleles ($K = 29.55$, $p = 5.48e^{-10}$, $LR = 38.50$), which means that positive selection is leading the microevolution at the high gastric cancer risk region of Colombia.

The Mestizo populations in the mountain zones in Colombia have an admixture of European and Amerind ancestries in a similar fashion that the bacterium mirroring the colonization process (*Kodaman et al., 2014*). When the Colombian *H. pylori* strains and human host ancestries are compared, a difference in the intensification of the disease aggressiveness is observed. Deleterious duplicated alleles of *alpA* were purged out of the population, but those strains with fixed *alpA* alleles under positive selection give advantages to the new types of strains in this region.

*H. pylori* is a bacterium that can display very fast local adaptive processes via mutation and recombination (*Cao et al., 2015*; *Furuta et al., 2015*). The phylogenetics, population genetics and protein evolutionary analysis suggest that *alpA* in Colombia has functionally divergent variants that are the result of gene conversion in a staggeringly short period

of time. AlpA proteins from the Colombian population show evidence of functional divergence, positive selection and episodic positive selection at specific sites. It is possible that the polymorphism of this adhesin in Colombia reflects the phylogeography and historical generation of Mestizos in Latin-America because *H. pylori* is a reliable biological marker of human migratory events (*Templeton, 2007*).

## CONCLUSION

The molecular evolution of virulence factors of *H. pylori* is currently gaining attention in the scientific community due to the new genomics and evolutionary findings around the world. Currently, Latin-American countries have emerged as evolutionary laboratories for *H. pylori*. To our knowledge, this study is the first study that presents statistically-supported evidence that *alpA* alleles from a high gastric cancer risk area from Colombia owe their variation patterns to gene conversion and purifying selection. In addition, a fast process of gene diversification followed by positive and relaxed selection has shaped three protein variants for the AlpA adhesin in Colombia.

## ACKNOWLEDGEMENTS

Thank to the anonymous reviewers for the constructive comments and guidelines.

### Funding

This work was supported by Colciencias Grant Contract No. 599-2014 to Gina Méndez and Andrés Julian Gutiérrez-Escobar and a Grant 41030610588 from Instituto Nacional de Cancerología to Maria Mercedes Bravo. The funders had no role in study design, data collection and analysis, decision to publish, or preparation of the manuscript.

### Grant Disclosures

The following grant information was disclosed by the authors:
Colciencias Grant Contract: 599-2014.
Instituto Nacional de Cancerología: 41030610588.

### Competing Interests

The authors declare there are no competing interests.

### Author Contributions

- Andrés Julián Gutiérrez-Escobar conceived and designed the experiments, performed the experiments, analyzed the data, contributed reagents/materials/analysis tools, prepared figures and/or tables, authored or reviewed drafts of the paper.
- Gina Méndez-Callejas analyzed the data, contributed reagents/materials/analysis tools, prepared figures and/or tables, authored or reviewed drafts of the paper, approved the final draft.

- Orlando Acevedo and Maria Mercedes Bravo analyzed the data, contributed reagents/materials/analysis tools, authored or reviewed drafts of the paper, approved the final draft.

## Data Availability

The sequences are provided as Supplemental Files.

## Supplemental Information

Supplemental information for this article can be found online at http://dx.doi.org/10.7717/peerj.4846#supplemental-information.

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
