# Peer review of "Rapid evolution of the Helicobacter pylori AlpA adhesin in a high gastric cancer risk region from Colombia"

_PeerJ, doi:10.7717/peerj.4846_

## Round 0.1 · original submission · Minor Revisions

· Academic Editor

Minor Revisions

Dear Dr. Gutierrez-Escobar and colleagues:

I have received two independent reviews of your manuscript, and both reviewers have raised concerns about several aspects of your work. I do not think addressing these issues is a "major" revision, but I am in agreement with their concerns and would like to see you address them accordingly. Please make sure you have a English expert read and edit your work. Also, please check the statistics and especially address Reviewer 2's concerns. Reviewer 1 raises an interesting concern about paralogy. Please respond to this, as your interpretation has major impact on the study design.

Thanks, and good luck with your revision!

-joe

Reviewer 1 ·

Basic reporting

The paper submitted Gutierrez-Escobar et., al. is a comprehensive evolutionary analysis of the AlpA gene in Colombian populations.
In general terms I find the paper well written. The state of the art and background of the paper is well supported, as well as the paper’s objective. However there are several gramatical errors that the authors must address. The ones that I’ve found are listed as follows:
37. I think is singular: “adhere”.
59. It progresses.
59. ulcers, not ulcera.
99,115: keep consistent nomenclature when referring to the DNA vs protein.
217: Sure is ten to the minus one? If that is the case use 0.2525 instead of the expression with exponent. That value is not significant and as a result the values of the GENCONV algorithm would not be significant as well. Please verify this.
Please show the lowest and the highest p-value.

In general, resolution of figures must be improved. Text in the phylogenetic trees and in figure 4 looks pixellated.

Experimental design

Rationale, experimental design, methodology and results are clear and within the scope of the journal.
Please include the original sequences used, the alignment used for this paper and sequence accession numbers if available. Please make sure all software include versions (i.e. muscle).

Validity of the findings

The nature of this paper is descriptive, there are no wet lab experiments carried out. Similar analyses have been done for other adhesins in H. pilory, however it is probably the first paper looking at Colombian data for this particular marker in this bacterium.
Its conclusions are within the reach of the experimental design. However, it is not clear to me why the authors mention gene duplication as a conclusión when apparently they didn’t use paralogous genes? AlpA belongs to a locus where other gene is located: AlpB which is very similar to AlpA. Both genes are paralogous to one another and the result of a duplication. Since the authors didn’t use any sequence belonging to AlpB no conclusión can be drawn with respect to gene duplication. All the analysis was done with AlpA, conclusions must be circunscribed within the reach of the experimental design and the data as well.

Reviewer 2 ·

Basic reporting

It’s better for the manuscript to be checked by a native English speaker or professional English editing service.

Experimental design

See comments below

Validity of the findings

See comments below

Additional comments

It’s better for the manuscript to be checked by a native English speaker or professional English editing service. For example, there are grammatical errors
L132: to identified -> to identify
L140: The HBK, Snn and chi squared were performed -> The HBK, Snn and chi squared tests were performed
L202: “The Tajima test was negative” -> “Tajima’s D was negative” or “Tajima’s D was negative”
L222: “Functional divergence … was performed” -> “Functional divergence analysis … was performed”?
L258: that -> than

L118, L131: reference of MEGA 7 should be
Kumar S Stecher G, Tamura K . 2016 . MEGA7: molecular evolutionary genetics analysis version 7.0 for bigger data sets. Mol Biol Evol . 33(7):1870–1874.

Figure 5 legend: p > 0.01 -> p < 0.01

Table 3: p > 0.05 -> p < 0.05? There is no explanation about the asterisks.

Table 1: There is no explanation what the Rm (table title) or RM (legend) is.

Figure 1 and 2 legends: remove “significant” because it is not defined in the manuscript and “significant consensus tree” is not a correct scientific term

L203-204: There is no explanation how Fisher’s exact test was conducted, and Table 2 doesn’t seem to show the result.

L206 and Table 3: There is no explanation what the Nm value is and how the values in Table 3 indicate a constant genetic flow.

L213-220: The number of recombination tracks depends on time to the most recent common ancestor of the Colombia strains. At least, it’s better to show when each strain was isolated if the authors have the data.

Table 4: There is no definition about the “Significant values”.

Overall, the authors don’t pay attention to multiple testing corrections when they repeat statistical tests and show p-values.

L131-133: Was the sliding windows analysis conducted using Mega 7? There is no result of the analysis shown in the manuscript.

2.5 and L222-233: The methods to infer positive selection used in the manuscript don’t assume recombination, and there is no theoretical justification to use them. A proper alternative is omegaMap.

Title: There is no explanation in the main text about “a high gastric cancer risk region from Colombia”. Where is it, and how is the cancer risk in the region?

---

## Round 0.2 · accepted · Accept

· Academic Editor

Accept

Dear Dr. Gutiérrez Escobar and colleagues:

Thanks for further revising your manuscript based on the concerns raised by the reviewers. I now believe that your manuscript is suitable for publication. Congratulations! I look forward to seeing this work in print, and I anticipate it being an important resource for the H. pylori and gastric cancer fields. Thanks again for choosing PeerJ to publish such important work.

Best,

-joe

#